# Association between Plant-Based Dietary Patterns and Risk of Cardiovascular Disease: A Systematic Review and Meta-Analysis of Prospective Cohort Studies

**DOI:** 10.3390/nu13113952

**Published:** 2021-11-05

**Authors:** Zuo Hua Gan, Huey Chiat Cheong, Yu-Kang Tu, Po-Hsiu Kuo

**Affiliations:** 1Institute of Epidemiology and Preventive Medicine, National Taiwan University, Taipei 100, Taiwan; frankie.gan@gmail.com (Z.H.G.); yukangtu@ntu.edu.tw (Y.-K.T.); 2Department of Family Medicine, Lianan Wellness Center, Taipei 105, Taiwan; 3Department of Internal Medicine, Dalin Tzu Chi Hospital, Chiayi County 622, Taiwan; hueychiat@gmail.com; 4Department of Psychiatry, National Taiwan University Hospital, Taipei 100, Taiwan

**Keywords:** cardiovascular disease, coronary heart disease, plant-based diet, meta-analysis

## Abstract

Plant-based diets, characterized by a higher consumption of plant foods and a lower consumption of animal foods, are associated with a favorable cardiovascular disease (CVD) risk, but evidence regarding the association between plant-based diets and CVD (including coronary heart disease (CHD) and stroke) incidence remain inconclusive. A literature search was conducted using the PubMed, EMBASE and Web of Science databases through December 2020 to identify prospective observational studies that examined the associations between plant-based diets and CVD incidence among adults. A systematic review and a meta-analysis using random effects models and dose–response analyses were performed. Ten studies describing nine unique cohorts were identified with a total of 698,707 participants (including 137,968 CVD, 41,162 CHD and 13,370 stroke events). Compared with the lowest adherence, the highest adherence to plant-based diets was associated with a lower risk of CVD (RR 0.84; 95% CI 0.79–0.89) and CHD (RR 0.88; 95% CI 0.81–0.94), but not of stroke (RR 0.87; 95% CI 0.73–1.03). Higher overall plant-based diet index (PDI) and healthful PDI scores were associated with a reduced CVD risk. These results support the claim that diets lower in animal foods and unhealthy plant foods, and higher in healthy plant foods are beneficial for CVD prevention. Protocol was published in PROSPERO (No. CRD42021223188).

## 1. Introduction

Cardiovascular disease (CVD), a leading cause of disease burden and deaths worldwide, is a group of disorders involving the heart and blood vessels, with coronary heart disease (CHD) and stroke being the two most common diseases in the group, each contributing 49.2% and 35.2% of annual CVD deaths, respectively [1]. Since an estimated 7.94 million annual CVD deaths and 188 million CVD related disability-adjusted life years (DALYs) can be attributed to poor diet alone, dietary interventions remain an important approach in the primary prevention of CVD [1]. 

Plant-based diets have received wide interest for their potential health benefits. Varying definitions of plant-based diets exist, but they are generally characterized by a lower consumption or avoidance of animal foods and a higher intake of plant foods. Vegetarian and vegan diets are the most restrictive, but plant-based diets may include eating patterns that are plant-dominant, while consuming some but fewer animal foods [2]. Although randomized controlled trials involving plant-based diets have been shown to improve cardiometabolic risk factors, atherosclerosis is a slowly progressing disease. Prospective observational studies are preferable to reflect the associations of long-term dietary patterns on CVD risk [3].

Vegetarians have been consistently associated with favorable CVD risk factors, such as lower levels of systolic and diastolic blood pressure, lower blood concentrations of total cholesterol and low-density lipoprotein cholesterol (LDL-C), and a decreased risk of type 2 diabetes mellitus [4,5,6]. Although two previous meta-analyses of observational studies came to the conclusion that vegetarians were associated with reduced CHD mortality, the association with total CVD and stroke were inconclusive [7,8]. Moreover, previous studies have mainly focused on mortality, with limited pooled evidence regarding the incidence of CVD, CHD, and stroke, which precedes mortality and may serve as a more informative outcome in primary prevention.

Prior observational studies have suffered from key limitations; they often compared individuals observing restrictive vegetarian or vegan diets with non-vegetarians. Since gradual dietary changes are easier to adopt, it is important to investigate how incremental reductions in animal foods and increases in plant foods affect CVD risk. Recent studies that used scoring indices to classify graded dietary adherence showed lower CVD or CHD incidence and mortality in individuals adhering to plant-based dietary patterns. In the Spanish PREDIMED (Prevención con Dieta Mediterránea) study, Martinez-Gonzalez et al. found that over a median follow-up of 5 years, individuals with the highest adherence to a pro-vegetarian diet score were associated with lower CVD mortality [9]. Satija et al. conducted a similar analysis of the Nurses’ Health Studies and the Health Professionals Follow-up Study in the US and found that the overall plant-based diet index (PDI) was inversely associated with incidences of CHD [10].

Given the possible benefits for the adoption of plant-based diets on lowering CVD risk, a quantitative assessment of the current research is warranted to provide conclusive evidence to inform clinical and public health recommendations. In addition, it is important to examine the association between vegetarian or plant-based diets on CHD and stroke separately since diet may affect them differently. In light of recent findings from several large prospective cohort studies, we conducted a systematic review and meta-analysis to address the knowledge gap regarding the associations between plant-based dietary patterns and total CVD, CHD, and stroke incidence. In addition, we investigated the dose–response associations of adherence to PDI patterns and the risk of total CVD, CHD, and stroke.

## 2. Materials and Methods

Findings from this systematic review and meta-analysis were reported according to the preferred reporting items for systematic review and meta-analysis (PRISMA) guidelines [11]. The protocol was published in the PROSPERO database (https://www.crd.york.ac.uk/PROSPERO/, accessed on 5 June 2021) with registration No. CRD42021223188.

### 2.1. Search Strategy

A systematic literature search for relevant studies was performed using the PubMed, EMBASE, and Web of Science databases up to December 2020. Search terms related to “plant-based diet”, “vegetarian”, “cardiovascular disease”, “coronary heart disease”, “stroke”, and “observational studies” were used (details are presented in the Appendix A). The search was restricted to human studies and studies published in English. We examined the reference lists of retrieved articles and recent reviews for potential publications. 

### 2.2. Study Selection

We included prospective studies examining the associations between adherence to plant-based dietary patterns and the incidence of cardiovascular disease among adults. Studies were considered eligible if they: (1) had a longitudinal cohort design (cohort, case-cohort or nested case-control) conducted in human adults aged 18 and above without CVD diseases during enrolment; (2) investigated the association between adherence to plant-based dietary patterns and the incidence of a composite or any cardiovascular disease (CVD), coronary heart disease (CHD), and stroke, and (3) provided data on relative risks with corresponding 95% confidence intervals of the outcomes. Unpublished studies, conference abstracts, and studies that used a cross-sectional design were excluded. For cohorts with more than 1 published study, we considered the availability of data for the analyses, and used the study with the largest population or longest follow up duration. Details of the inclusion and criteria are provided in Appendix A.

### 2.3. Data Extraction and Quality Assessment

Two authors (ZHG and HCC) screened the retrieved articles independently. Any discrepancies were resolved with a group discussion with a third investigator (PHK). ZHG and HCC extracted the following information from each study: first author and publication year of the article, cohort name, region, number of participants and cases, mean age, mean body mass index (BMI), duration of follow-up, sex composition (percentage of women), dietary assessment and classification, method of case ascertainment, percentage of current smokers, and covariates that were included in the statistical models. Quality assessment for the individual studies was examined using the Newcastle–Ottawa criteria for nonrandomized studies [12], which assigns a maximum of 9 points to each study, including 4 points for selection, 2 points for comparability, and 3 points for the assessment of outcomes. Studies with scores of 0–3, 4–6, and 7–9 were identified as low, moderate, and high quality, respectively.

### 2.4. Methods Used to Assess the Adherence to Plant-Based Dietary Patterns

The definition of a plant-based diet varied according to the studies, but it was generally defined as a diet involving a higher consumption of plant foods and either a lower consumption or complete avoidance of animal foods [2,13]. The studies used the following methods to classify plant-based dietary patterns: (1) a questionnaire about the avoidance of animal foods; (2) a posteriori factor analysis to derive dietary patterns; and (3) a priori plant-based dietary pattern index scores.

There were three main plant-based dietary pattern scoring methods used in the cohort studies. In all of these scoring systems, the lowest intake quintile of plant food items receives 1 point, while the top quintile receives 5 points. Animal food items are reversely coded so that the lowest quintile receives 5 points, and the highest quintile receives 1 point. (1) The Provegetarian diet index (PV) and overall plant-based index (oPDI) were calculated with the above methods, with higher scores reflecting a higher intake of plant foods and a lower intake of animal foods. (2) The healthful plant-based index (hPDI) positively scores healthy plant food components (such as unrefined grains, fruits, vegetables, nuts, and legumes), while reversely scoring the intake of unhealthy plant foods (such as refined grains, sugar sweetened beverages, sweets, and desserts) and animal foods. (3) The unhealthful plant-based index (uPDI) positively scores unhealthy plant foods while reversely scoring the intake of healthy plant foods and animal foods. Depending on the numbers of food items included in each study, the plant-based indices may have a minimum score ranging from 11 to 18, and a maximum score ranging from 55 to 90. 

For studies that used dietary scores or a posteriori-defined methods to categorize plant-based dietary pattern adherence, we used the risk estimate that compared the highest to the lowest adherence category. For the other studies, we included risk estimates comparing the diet category that represents the greatest restriction of animal foods (vegan, vegetarian or pesco-vegetarian) with the least restrictive diet category (omnivorous or non-vegetarian).

### 2.5. Data Synthesis and Analysis

To investigate the relationship between plant-based diets and CVD, CHD, and stroke, we summarized the risk estimate for the highest versus the lowest adherence to plant-based dietary patterns using odd ratios, relative risks (RR), and hazard ratios (along with 95% confidence intervals, CI) for the included studies [14]. Studies were grouped according to the different clinical outcomes, which included: (1) CVD incidence, a composite of any fatal or nonfatal CVD, CHD, or stroke event; (2) CHD incidence; and (3) stroke incidence (including total, ischemic, and hemorrhagic stroke). Risk estimates given by statistical models with full adjustment of confounding variables were chosen for the meta-analyses. The DerSimonian–Laird random effects models were used to account for variations in the designs of the included studies. We used I^2^ and Cochran’s Q statistics to quantify heterogeneity. Subgroup analyses were performed to investigate potential sources of heterogeneity, stratified by sex, country, follow up duration, mean age at baseline, methods of dietary classification, exposure ascertainment, and study quality. Meta-regression was also undertaken for continuous variables of the study characteristics, including the mean age and BMI at baseline, the proportion of smokers, and the length of follow up. 

A leave-one-out sensitivity analysis was performed by iteratively removing one study at a time to examine the influence of a single study on the overall effect. Publication bias was examined by visually inspecting funnel plots and, if there was a sufficient number of studies, with the Egger and Begg regression tests. A trim-and-fill method was adopted to detect the effect of publication bias on the overall result [15].

### 2.6. Data Harmonization and Dose–Response Meta-Analyses

In studies that defined adherence to plant-based dietary patterns using plant-based diet indices, index scores were transformed to a 0–100 scale, with a higher score reflecting a higher adherence to the dietary index. The mean or median score of each plant-based diet index category and related RR and 95% CI were extracted. We used the generalized least squares method described by Greenland and Longnecker for the dose-response meta-analysis [16]. To explore the potential nonlinear association between plant-based dietary index scores and cardiovascular disease risk, a two-stage random effects meta-analysis model was performed by first fitting the restricted cubic splines model with nodes at fixed centiles of 5%, 35%, 65%, and 95% of the distribution of the plant-based dietary index scores. A linear dose–response association was performed to estimate RRs for a 25% increment in plant-based diet index scores with cardiovascular disease risk. A likelihood ratio test was used to assess the difference in the model fits between the nonlinear and linear models, with *p* < 0.05 indicating a better model fit for the nonlinear model.

All statistical analyses were performed with the package dosresmeta [17] for R Statistical Software, version 3.5.1 (Foundation for Statistical Computing, Vienna, Austria). Reported probabilities (*p* values) were two-sided, with *p* < 0.05 considered statistically significant.

## 3. Results

### 3.1. Literature Search

In total, 923 articles were identified in the search. After excluding duplicate papers and those that did not meet the inclusion criteria, 122 full text articles were reviewed, and 10 articles describing 9 separate cohorts were included in our systematic review and meta-analysis (Figure 1). 

### 3.2. Characteristics of the Included Studies

The general characteristics of the included studies are presented in Table 1, which comprised a total of 698,707 study participants with 137,968 cases of incident cardiovascular disease and a follow up duration ranging from 5 to 36 years. All studies used a prospective cohort design. The mean age of the study participants at baseline ranged from 36 to 64 years. In total, five publications came from the United States, four were from the United Kingdom, and one was from Taiwan.

Of the included publications, some studies were conducted on the same study populations. The studies by Crowe et al. [18] and Tong et al. [19] were performed on the European Prospective Investigation into Cancer and Nutrition (EPIC)-Oxford cohort. The Tong et al. [19] study was included because it had a larger number of participants with a longer duration of follow up. Studies by Satija et al. [10] and Shan et al. [20] were conducted on the Nurses’ Health Study (NHS), Nurses’ Health Study II (NHS2), and Health Professionals Follow-up Study (HPFS) datasets, while studies by Heianza et al. [21] and Petermann-Rocha et al. [22] analyzed the UK Biobank cohort. Shan et al. [20] and Petermann-Rocha et al. [22] were included in the main analysis because of their larger sample sizes. However, Satija et al. [10] and Heianza et al. [21] were included in the dose-response analysis because of the availability of plant-based index scores data. Studies by Judd et al. [23] and Shikany et al. [24] were from the same cohort (Reasons for Geographic and Racial Differences in Stroke, REGARDS) study but reported different outcomes and were included in separate analyses.

One study used a web-based 24 h dietary recall [21], while all others used a validated food frequency questionnaire to assess diet. Four studies used plant-based dietary indices to characterize adherence to plant-based dietary patterns. Four studies compared study participants following a vegetarian or vegan dietary pattern with non-vegetarians, while two studies derived a plant-based dietary pattern using a factor analysis approach. In all studies, incident cardiovascular disease was identified through either medical and death record linkage, or through self-report with medical record confirmation. All studies adjusted for conventional cardiovascular risk factors, including age, sex, BMI, physical activity, and smoking status. Most studies further adjusted for alcohol consumption, energy intake, menopause status (in females), personal history of hypertension, dyslipidemia, and type 2 diabetes mellitus (Table 1).

### 3.3. Plant-Based Dietary Pattern and Risk of Cardiovascular Disease

Six publications from nine unique cohorts examined the association of plant-based diets and incident cardiovascular disease, with a total of 698,707 study participants and 137,968 cases of CVD. When comparing the highest versus lowest adherence categories, the plant-based dietary pattern was associated with a lower risk of CVD (RR 0.84; 95% CI 0.79 to 0.89; *p* < 0.05). There was significant heterogeneity among the included studies (I^2^ = 65%, *p* < 0.01 for heterogeneity). The forest plot of the pooled RRs is presented in Figure 2.

### 3.4. Plant-Based Dietary Pattern and Risk of Coronary Heart Disease

Five publications describing six separate cohorts examined the association between a plant-based dietary pattern and risk of CHD. These studies included a total of 694,191 participants and 36,781 CHD events. There was an inverse association between the highest and lowest adherence of plant-based dietary patterns and CHD (RR 0.89; 95% CI 0.81 to 0.97; *p* < 0.05). No significant heterogeneity was seen between the studies (I^2^ = 48%, *p* = 0.09 for heterogeneity, Figure 3).

### 3.5. Plant-Based Dietary Pattern and Risk of Stroke

Five publications from eight separate cohorts, with a total of 720,926 participants and 13,370 events, examined the association between a plant-based dietary pattern and risk of stroke. The summary effect from the pooled studies was RR 0.87; 95% CI 0.73 to 1.03; *p* = 0.11, suggesting an insignificant trend for the protective effect from the highest versus the lowest adherence to the plant-based dietary pattern. However, there was significant heterogeneity between the included studies (I^2^ = 76%, *p* < 0.01 for heterogeneity, Figure 4). Only two studies reported separate analyses on ischemic and hemorrhagic stroke subtypes. The forest plot is presented in Appendix A.

### 3.6. Dose-Response Meta-Analysis

Five separate cohorts from three studies that assessed dietary patterns using plant-based dietary indices were included in the dose–response analyses. The UK Biobank study only reported the healthful PDI scores while the overall, healthful, and unhealthful PDIs data were available from the NHS, NHS2, HPFS and ARIC (Atherosclerosis Risk in Communities Study) cohorts. 

The dose-response associations of the overall, healthful, and unhealthful PDIs with CVD risk were, in general, linear, with an inverse relationship between the overall and healthful PDIs and the incident CVD and with a positive relationship between the unhealthful PDI and the CVD risk (Figure 5). Based on the linear dose-response analysis, each additional 25% increase in the overall PDI and healthful PDI scores was associated with a 15% (RR: 0.85, 95% CI: 0.80 to 0.90) and 16% (RR: 0.84, 95% CI: 0.75 to 0.94) reduction in CVD risk, respectively. An unhealthful PDI was significantly associated with a higher CVD risk (RR: 1.13 (95% CI 1.02 to 1.26) per 25% increase in the unhealthful PDI). The differences in the model fit between the linear and nonlinear models were all non-significant, suggesting linear relationships between PDIs and CVD risk.

### 3.7. Subgroup Analysis, Meta-Regression, and Assessment of Publication Bias

A series of subgroup analyses were conducted based on our predefined criteria (Appendix A). We found that the mean age at baseline, outcome definition, study region, and study quality were significant sources of heterogeneity for the association between plant-based dietary patterns and the risk of CVD, CHD, and stroke. Despite the existence of heterogeneity, the direction of association and the significant findings on the adherence to plant-based dietary patterns and CVD and CHD risks were generally consistent across different strata. The meta-regression analyses revealed that the mean age of the study participants at baseline was a significant source of heterogeneity for the association between plant-based diets and CVD and CHD risks (Appendix A).

We detected two studies with small sample sizes (TCHS and TCVS) that reported a substantial effect size but updated trim-and-fill analyses with imputed studies did not alter the results, suggesting no significant publication bias (Appendix A). Egger and Begg regression tests were not performed due to the limited number of included studies. The quality assessment of the included studies using the Newcastle–Ottawa criteria is detailed in Appendix A. All studies were deemed to be high quality, with common weaknesses being the lack of representativeness of the cohort and the failure to report the proportion of complete follow up in the cohort. Sensitivity analysis using the leave-one-out method showed that the exclusion of any single study from the analyses did not appreciably alter the pooled effect sizes for the associations of plant-based diets with the risk of CVD and CHD (Appendix A). However, the association between plant-based diets and stroke did achieve statistical significance after excluding Tong et al. [19].

## 4. Discussion

In this systematic review and meta-analysis, we found that plant-based diets are associated with a lower risk of incident CVD and CHD. The results were generally significant across different subgroups and were robust in our sensitivity analyses. In studies that used a graded adherence approach to classify plant-based diets, we identified a significant dose–response association between adherence to the overall PDI and healthful PDI with lower CVD risk. On the contrary, greater adherence to the unhealthful PDI was associated with a higher CVD risk. Collectively, these findings support the adoption of a healthy plant-based diet for the primary prevention of CVD and CHD.

Our findings are broadly consistent with studies on other plant-based dietary patterns that do not restrict animal food consumption. Previous meta-analyses of observational studies revealed that highest adherence to the Mediterranean and DASH (Dietary Approaches to Stop Hypertension) diets, which emphasize a higher consumption of plant foods, is associated with lower CVD risks, with pooled effect sizes of RR 0.81 (95% CI 0.74–0.88) and RR 0.80 (95% CI 0.76–0.85), respectively [27,28], which are similar to our findings. This is in line with recommendations issued by the American Heart Association/American College of Cardiology (AHA/ACC) which include vegetarian, Mediterranean, and DASH diets as dietary patterns that meet heart healthy guidelines [29].

Our results showed similar protective effects for CVD risk with adherence to the overall PDI and healthful PDI. Since roughly 50% of global CVD cases are comprised of CHD, the risk reduction in CVD may be largely attributable to CHD. In the Nurses’ Health Study 2 and Health Professionals Follow-up Study, Wang and colleagues observed that the replacement of animal saturated fats with polyunsaturated fats resulted in a lower CHD risk [30]. On the other hand, a meta-analysis of 123 studies showed that a higher consumption of red meat and processed meat was linearly associated with increased CHD risks, RR 1.15 (95% CI 1.08–1.23) and RR 1.27 (95% CI 1.09–1.49), respectively, while egg consumption was associated with increased heart failure risk, RR 1.16 (95% CI 1.03–1.31) [31]. These findings suggest that the avoidance or lower consumption of animal foods may be necessary to obtain the heart healthy effects of plant-based diets. The dose–response association further suggests that even without a strict restriction of animal foods, a gradual shift from animal foods to healthy plant foods will likely lead to cardioprotective benefits.

Adherence to the unhealthful PDI is associated with a higher CVD risk in a linear fashion. A study conducted on the French NutriNet-Santé cohort found that the intake of ultra-processed food was associated with a higher risk of CVD (for each 10% increment of intake, HR 1.12 (95% CI 1.05–1.20)) [32]. The result remained statistically significant after adjustment for several markers of the nutritional quality of the diet including a healthy dietary pattern derived by a principal component analysis. In a cross-sectional analysis of the Nurses’ Health Study 2, a higher unhealthful PDI was associated with higher concentrations of leptin and insulin (4.4% and 4.8%, respectively; *p* ≤ 0.05), which are biomarkers that are both associated with a higher cardiometabolic risk [33]. Current evidence along with our findings suggest that even with a lower consumption of animal foods, a higher intake of processed plant foods may be detrimental to cardiovascular health. 

In the meta-regression analyses, we observed a significantly greater protective effect of plant-based diets on CVD and CHD risks in studies conducted on participants that were younger during recruitment. Similarly, stronger inverse associations in younger participants have been observed in previous studies investigating the association between vegetarian diets and CHD mortality [34]. A recent meta-analysis concluded that combined healthy lifestyle factors are more beneficial for reducing CVD in younger adults [35]. One possible explanation is that since atherosclerotic changes precede CVD for decades, greater benefits may occur in individuals that adhere to heart healthy diets earlier in life. In our subgroup analysis, heterogeneity decreased when CHD and stroke were considered separately instead of being pooled together as CVD. This supports our hypothesis that plant-based diets affect the risks of these two diseases differently and should be examined separately.

We observed a protective trend for stroke not reaching statistical significance with adherence to a plant-based diet and found that this study region is a significant source of heterogeneity. According to previous research, there have been inconsistent findings regarding the effects of various dietary and CVD modifiable factors on stroke. For example, vegetarian diets were associated with lower LDL-C [5], but the China Kadoorie Biobank study revealed that there was a higher risk of hemorrhagic stroke but a lower risk of ischemic stroke with lower LDL-C [36]. INTERSTROKE, a multi-regional case-control study in 32 countries identified regional variations in the relative importance of diet and other modifiable factors for stroke risk [37]. Two studies performed in the UK and Taiwan included in our analysis reported different findings on the association of vegetarian diets on stroke subtypes [19,26]. Stroke is a heterogeneous class of disease and there are limited data to compare different stroke subtypes, ethnicity, and regions. Consequently, the association between plant-based diets and stroke remains inconclusive, and more studies with greater numbers of accumulated events are required to examine this association.

### 4.1. Potential Mechanisms

Plant-based diets are generally high in the intake of whole grains, fruits, vegetables, nuts, and legumes, all of which have been associated with cardiovascular health [31]. These plant foods are low in energy density but high in fiber, vitamins, minerals, and phytonutrients. They have been shown to be associated with favorable weight [38], glycemic [6,39] lipid [5], blood pressure [4], and inflammation outcomes [40] in clinical and observational trials, all of which are involved in the atherosclerotic process. Plant-based diets are also low in saturated fat, which may lead to lower serum cholesterol levels. In a meta-analysis, long term trials that reduced dietary saturated fats reduced the risk of combined cardiovascular events by 21% [41]. On the other hand, several animal food groups have been consistently associated with increased CVD and its risk factors [31,42,43]. This is possibly due to the high heme iron content in animal meats which may lead to inflammation, contributing to the onset and progression of arteriosclerosis [44]. Choline and L-carnitine from animal foods are also metabolized by gut microbes to generate trimethylamine N-oxide, a metabolite that promotes atherosclerosis and is associated with increased CVD risks [45,46].

### 4.2. Strengths and Limitations

Our meta-analysis has several strengths. We included prospective studies with large numbers of participants and events, the majority of which had long periods of follow up. These studies utilized medical or death records as case ascertainment methods, lowering the risk of misclassification in the outcome assessments. We were able to conduct a dose-response analysis, assessing the linear and non-linear associations between a graded plant-based diet adherence and CVD risks. Lastly, to test the robustness of the results, several analytic methods including subgroup stratification and sensitivity analysis were performed.

However, it should be noted that this study presents several common limitations known to meta-analyses and observational studies. Due to the observational nature of the included studies, there may have been residual and unmeasured confounders that we could not resolve. Due to limited available subgroup data, only some of the potential sources of heterogeneity were identified. However, random effects analyses were performed to compensate for this limitation. In addition, since there were wide differences in the methods of dietary assessment across the studies; therefore, measurement errors and misclassifications could not be ruled out. Most studies assessed diet during enrolment and this may not have reflected the association between subsequent dietary changes and CVD risks. Furthermore, the included studies were from high-income countries and the results may not be generalizable to other low- or middle-income countries where plant-based dietary patterns may differ considerably.

## 5. Conclusions

In this study, we found that plant-based diets were associated with a lower risk of incident CVD and CHD. While adherence to a healthy plant-based diet is likely to lower the risk of CVD, adherence to an unhealthy plant-based diet may increase the risk of CVD. These findings have important public health implications, as consuming less animal foods and unhealthy plant foods and increasing the intake of healthy plant foods could have significant benefits on cardiovascular health.

## Figures and Tables

**Figure 1 nutrients-13-03952-f001:**
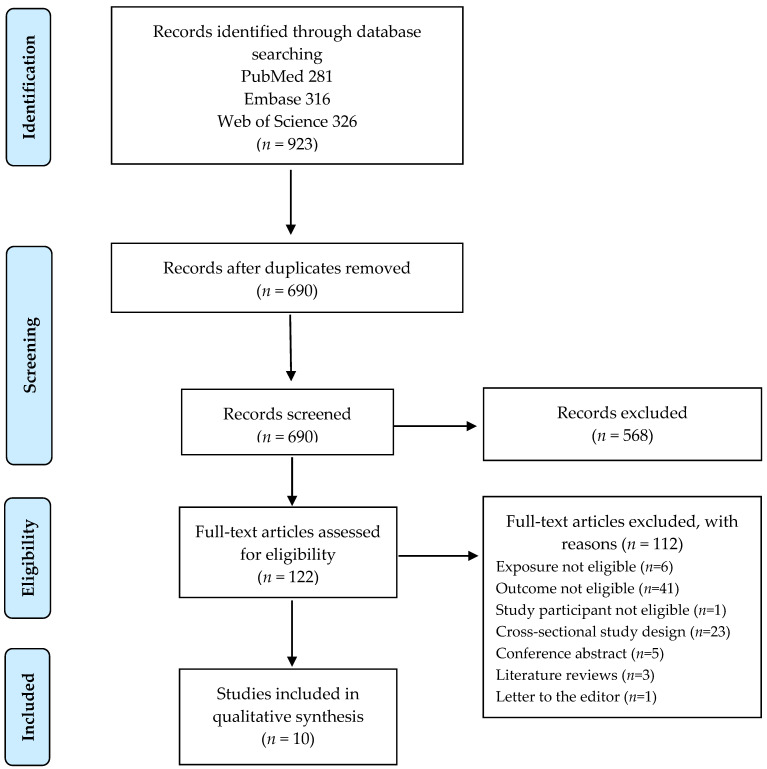
Flow diagram of the literature search.

**Figure 2 nutrients-13-03952-f002:**
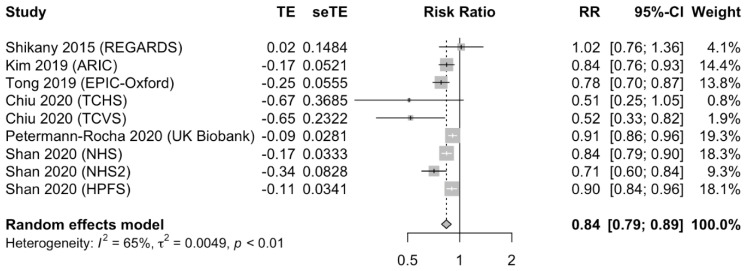
Forest plot of the adjusted relative risk (RR) of cardiovascular disease for the highest versus the lowest adherence to plant-based dietary patterns. Pooled risk estimates and 95% confidence intervals (CI) using a random effects model for meta-analysis are in bold. Abbreviations: ARIC, Atherosclerosis Risk in Communities Study; CI, confidence interval; EPIC, European Prospective Investigation into Cancer and Nutrition; NHS, Nurses’ Health Study, NHS2, Nurses’ Health Study II; REGARDS, Reasons for Geographic and Racial Differences in Stroke; RR, relative risk; seTE, standard error of treatment effect; TCHS, Tzuchi Health Study; TCVS, Tzuchi Vegetarian Study; TE, treatment effect; UK, United Kingdom.

**Figure 3 nutrients-13-03952-f003:**
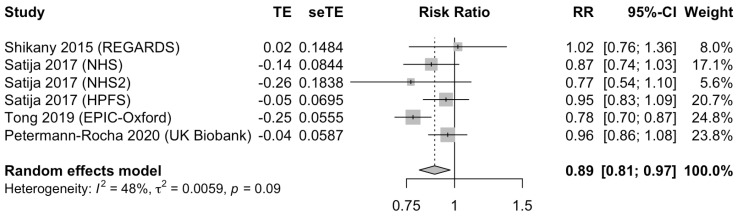
Forest plot of the adjusted relative risk (RR) of coronary heart disease for the highest versus the lowest adherence to plant-based dietary patterns. Pooled risk estimates and 95% confidence intervals (CI) using a random effects model for meta-analysis are in bold. Abbreviations: CI, confidence interval; EPIC, European Prospective Investigation into Cancer and Nutrition; NHS, Nurses’ Health Study, NHS2, Nurses’ Health Study II; REGARDS, Reasons for Geographic and Racial Differences in Stroke; RR, relative risk; seTE, standard error of treatment effect; TE, treatment effect; UK, United Kingdom.

**Figure 4 nutrients-13-03952-f004:**
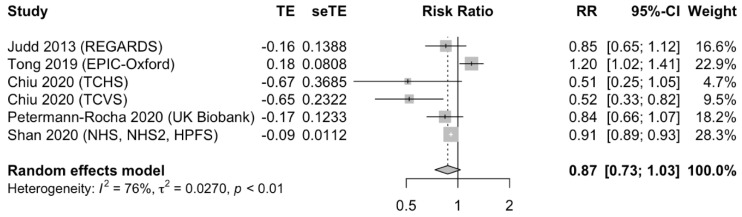
Forest plot of the adjusted relative risk (RR) of stroke for the highest versus the lowest adherence to plant-based dietary patterns. Pooled risk estimates and 95% confidence intervals (CI) using a random effects model for meta-analysis are in bold. Abbreviations: CI, confidence interval; EPIC, European Prospective Investigation into Cancer and Nutrition; NHS, Nurses’ Health Study, NHS2, Nurses’ Health Study II; REGARDS, Reasons for Geographic and Racial Differences in Stroke; RR, relative risk; seTE, standard error of treatment effect; TCHS, Tzuchi Health Study; TCVS, Tzuchi Vegetarian Study; TE, treatment effect; UK, United Kingdom.

**Figure 5 nutrients-13-03952-f005:**
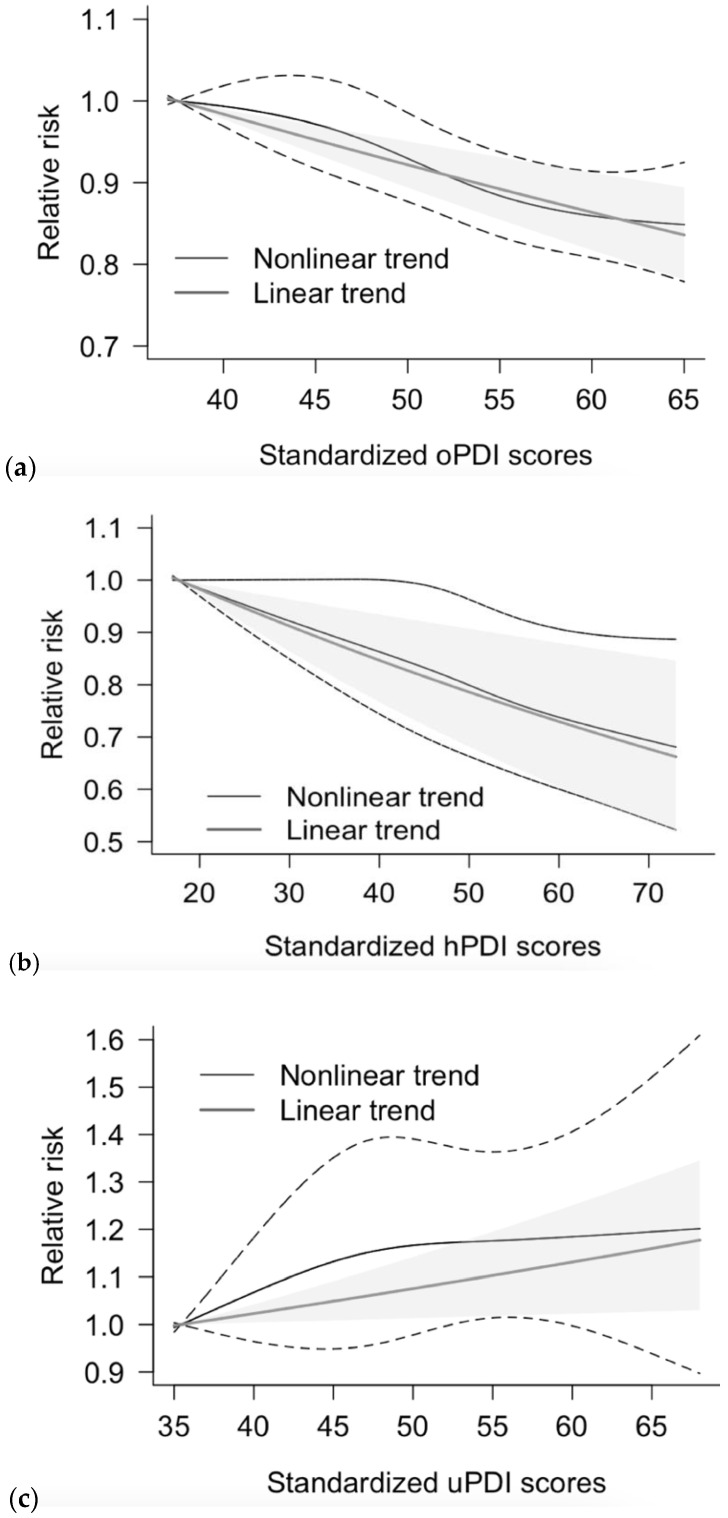
Dose–response analyses for the potential linear and nonlinear associations between plant-based diet indices (PDI) and incident cardiovascular disease (CVD). The shaded areas represent the 95% confidence intervals (CI) for the fitted linear trend (gray solid lines). The dashed line areas represent the 95% confidence intervals (CI) for the fitted nonlinear trend (black solid lines). (**a**). Overall PDI was associated with a lower risk of CVD in a linear fashion (RR: 0.85 (95% CI 0.80 to 0.90) per 25% increase, *p* for nonlinearity <0.01; *p* for significance of the curve = 0.53; *p* for linear association < 0.01). (**b**). Healthful PDI was associated with a lower risk of CVD in a linear fashion (RR: 0.84 (95% CI 0.75 to 0.94) per 25% increase, *p* for nonlinearity = 0.01; *p* for significance of the curve = 0.97; *p* for linear association < 0.01). (**c**). Unhealthful PDI was associated with a higher risk of CVD in a linear fashion (RR: 1.13 (95% CI 1.02 to 1.26) per 25% increase, *p* for nonlinearity < 0.01; *p* for significance of the curve = 0.13; *p* for linear association < 0.01).

**Table 1 nutrients-13-03952-t001:** The baseline characteristics of the included studies on plant-based diet and the incidence of cardiovascular disease.

Reference	Country	Cohort	Sex	Mean Age, Years	Mean F/U, Years	*n*/N	Exposures	Dietary Assessment	Outcome	Outcome Ascertainment	Adjustments for Confounders
Crowe 2013 [18]	UK	EPIC-Oxford	M/F	44	11.6	1235/44,561	Vegetarians vs. non-vegetarians	Questionnaire inquiring about meat and fish avoidance	CHD	Medical or death record	Sex, method of recruitment, region of residence, age, smoking, alcohol, physical activity, educational level, Townsend Deprivation Index, use of oral contraceptives or hormone therapy, and BMI
Judd 2013 [23]	USA	REGARDS	M/F	64	5.7	490/28,151	Plant-based dietary pattern adherence comparing extreme quartiles	Validated 110-item FFQ, plant-based dietary pattern derived from factor analysis	Stroke	Self-report and medical records confirmation	Age, race, region, sex, age-race, income, education, total energy, smoking, and sedentary behavior
Shikany 2015 [24]	USA	REGARDS	M/F	64	5.8	536/17,418	Plant-based dietary pattern adherence comparing extreme quartiles	Validated 110-item FFQ, plant-based dietary pattern derived from factor analysis	CHD	Self-report and medical records confirmation.	Age, sex, race, age–race interaction, education, household income, region, total energy intake, smoking, physical activity, body mass index, waist circumference, and history of hypertension, dyslipidemia, and diabetes mellitus.
Satija 2017 [10]	USA	NHS	F	54	36	3233/73,710	oPDI, hPDI, and uPDI, comparing extreme deciles	Validated 131-item FFQ, plant-based dietary indices from 18 food groups	CHD	Self-report and medical or death records confirmation	Age, smoking status, physical activity, alcohol intake, multivitamin use; aspirin use, family history of CHD, margarine intake, energy intake, baseline hypertension, hypercholesterolemia, and diabetes, and updated body mass index. Adjusted also for post-menopausal hormone use in NHS and NHS2, and for oral contraceptive use in NHS2
		NHS2	F	55	24	667/93,329					
		HPFS	M	53	26	4731/43,259					
Kim 2019 [25]	USA	ARIC	M/F	54	25	4381/12,168	oPDI, hPDI and uPDI, comparing extreme deciles	Validated 66-item FFQ, plant-based dietary indices from 17 food groups	CVD	Self-report and medical or death records confirmation.	Age, sex, race–center, total energy intake, education, smoking status, physical activity, alcohol consumption, and margarine consumption
Tong 2019 [19]	UK	EPIC-Oxford	M/F	45	18.1	2820/48,188	Vegetarians vs. meat eaters	Questionnaire inquiring about consumption of meat, fish, dairy products, and eggs	CHD Stroke	ICD-9 and ICD-10 codes by record linkage	Age, sex, method of recruitment, region, year of recruitment, education, Townsend deprivation index, smoking, alcohol consumption, physical activity, dietary supplement use, oral contraceptive, and hormone replacement therapy use in women
						1072/48,188					
Chiu 2020 [26]	Taiwan	TCHS	M/F	52	10	54/5050	Vegetarians vs. non-vegetarians	Questionnaire inquiring about meat and fish avoidance	Stroke	ICD-9 codes by record linkage	Sex, smoking, alcohol drinking, betel nut, leisure time, physical activities, education, hypertension, diabetes mellitus, dyslipidemia, ischemic heart disease, and body mass index
		TCVS	M/F	49	10	121/8302					
Heianza 2020 [21]	UK	UK Biobank	M/F	56	5	1812/156,148	hPDI comparing extreme quintiles	Web-based 24 h dietary assessment, plant-based dietary indices from 17 food groups	CVD CHD Stroke	ICD-9 and ICD-10 codes by record linkage	Age, sex, ethnicity, education, parental history of heart disease, smoking habit, physical activity, multivitamin use, total energy intake, alcohol consumption, Townsend Deprivation Index, BMI, hypertension, dyslipidemia, and type 2 diabetes
						1162/156,148					
						697/156,148					
Petermann-Rocha 2020 [22]	UK	UK Biobank	M/F	56.48	8.5	106,690/398,448	Vegetarians vs. meat-eaters	Questionnaire inquiring about consumption of dairy, fish, meat, and poultry	CVD CHD Stroke	ICD-10 codes by record linkage	Age, sex, deprivation, ethnicity, comorbidities, smoking, alcohol intake, total sedentary time, physical activity, and body mass index
						24,794/418,287					
						5946/422,102					
Shan 2020 [20]	USA	NHS, NHS2, HPFS	M/F	53.2	32	23,366/209,133	hPDI comparing extreme quintiles	Validated 131-item FFQ, plant-based dietary indices from 18 food groups	CVD CHD Stroke	Self-report and medical or death records confirmation	Age, race/ethnicity, body mass index, physical activity, smoking, status, alcohol intake, menopausal status, oral contraceptive use, marital status, alone or with others, family history of myocardial infarction, total energy intake, multivitamin use, and aspirin use
						18,092/209,133					
						5687/209,133					

Abbreviations: ARIC, Atherosclerosis Risk in Communities Study; BMI, body mass index; CHD, coronary heart disease; CVD, cardiovascular disease; EPIC, European Prospective Investigation into Cancer and Nutrition; F, Female; FFQ, Food Frequency Questionnaire; HPFS, Health Professionals Follow-up Study; hPDI, healthful plant-based dietary index; ICD, International Classification of Disease; M, Male; NHS, Nurses’ Health Study, NHS2, Nurses’ Health Study II; oPDI, overall plant-based dietary index; REGARDS, Reasons for Geographic and Racial Differences in Stroke; TCHS, Tzuchi Health Study; TCVS, Tzuchi Vegetarian Study; uPDI, unhealthful plant-based dietary index; UK, United Kingdom; USA, United States of America.

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
