# Peer review of "Association between Plant-Based Dietary Patterns and Risk of Cardiovascular Disease: A Systematic Review and Meta-Analysis of Prospective Cohort Studies"

_nutrients, 2021, doi:10.3390/nu13113952_

Round 1

Reviewer 1 Report

Zuo Hua Ga et al. address the important topic of plant-based diets and the risk of CVD. The authors have concluded results from 9/10 prospective studies with almost 700,000 observations. Overall, this is impressive, important, and interesting work that shows that plant-based diets are associated with a lower risk of incidents of CVD and CHD. I have very few suggestions for the authors:

  1.  Given the differences in follow-up time for each cohort, ranging from 5 to 36 years, it seems that a sub-analysis of all cohorts below and above the median time of follow-up is needed. 
  2. Supplemental figure 1 should be figure 1 in the main text. Also - the abstract mentions 9 studies included in this work, but the figure and the text itself include 10 studies. Could you clarify?

Author Response

Response to Reviewer 1 Comments

Thank you for the feedback, we appreciate your time in helping us improve our manuscript.

Point 1: Given the differences in follow-up time for each cohort, ranging from 5 to 36 years, it seems that a sub-analysis of all cohorts below and above the median time of follow-up is needed.

Response 1: We did find that length of follow up was not a significant source of heterogeneity in both the subgroup and meta-regression analyses. We previously only included the results of the meta-regression analysis, but we have also included the results of the subgroup analysis in the updated manuscript (Supplemental Tables 3 and 4).

Point 2: Supplemental figure 1 should be figure 1 in the main text. Also - the abstract mentions 9 studies included in this work, but the figure and the text itself include 10 studies. Could you clarify?

Response 2: Supplemental Figure 1 was changed to Figure 1. There were 10 studies (publications) describing 9 different cohorts. Some studies were conducted on the same study populations, while some publications included multiple cohorts. We updated the description in the Abstract and Results to better clarify this.

Reviewer 2 Report

Dear authors,

thx for the opportunity to do this interesting and meaningful review, and happy in helping to furhter improve an intersting review;

however, some points have to be addressed for consideration of publication:

  • Intro must be improved and gap as well as aim of this paper has to be clearly addressed (what is the different to similar reviews on this or familiar topics from the recent 2 years, there have been some -> see also: Novelty/originality -> you have to better "sell" your USB of this spedific paper)
  • add to clarify methods
  • Results: already good presentation of results, I miss more focused presentation related to the aim of study and especially 1-2-3 research questions: what are these Research questions in detail and could they be completely answered with the results presented
  • Significance/Sound theory (body of science:; intro vs. discussion): clarify content, goal. reseach Questions, Methods, reults and take home message, 

I am convinced with a focused 2nd round this paper will be much improved, although starting at a good level.

All good success

Author Response

Response to Reviewer 2 Comments

Thank you, we appreciate your comments and we believe that it will help us further improve our manuscript.

Point 1: This work is not only a literature search but a systematic review and should be identified as such in the abstract. No details on inclusion/exclusion criteria provided. No details provided on risk of bias assessment. No brief summary on limitations provided. Source of funding not indicated. Register and register number not indicated.

Response 1: Being mindful of the limited words allowed (200 words in the abstract), we updated the abstract the best we can, to both highlight the main methods and results of this study, and to give relevant information regarding inclusion/exclusion criteria and registration number. Risk of bias and limitations of the study were included in the main text due to word limitations.

Point 2: What the background is lacking is an overview of randomized controlled trial (RCT) evidence in the area of the research investigation, as such evidence has the lowest risk of bias, followed by quasi-experimental evidence, and then observational evidence. The rationale is missing for why this review does not include RCT data or even experimental data but focuses entirely on observational data.

Response 2: We updated the manuscript to clarify our rationale of including observational studies. On page 1, we stated “Although randomized controlled trials involving plant-based diets have been shown to improve cardiometabolic risk factors, atherosclerosis is a slowly progressing disease. Prospective observational studies are preferable to reflect the associations of long-term dietary patterns on CVD risk.[3].”

Point 3: So, what is the goal after identifying the gap in the literature? Clearly outline what the aim of this specific review is. is NOT the same as describing in 1 sentence what you did

Response 3: We revised our introduction to better reflect our study aim. On page 2, we stated “Given the possible benefits and low risk for the adoption of plant-based diets on CVD prevention, a quantitative assessment of current research is warranted to provide conclusive evidence in informing clinical and public health recommendations. In addition, it is important to examine the association of vegetarian or plant-based diets on CHD and stroke separately since diet may affect them differently. In light of recent findings from several large prospective cohort studies, we conducted a systematic review and meta-analysis to address the knowledge gap regarding the associations between plant-based dietary patterns and total CVD, CHD and stroke incidence. In addition, we investigated the dose-response associations of adherence to PDI patterns and the risk of total CVD, CHD and stroke.”

Point 4: Population unclear: no age provided and no clear indication of baseline health status.

Response 4: In our literature search, we included studies with a population of healthy adults, excluding children and adults with previous cardiovascular disease diagnoses (Supplemental Table 2). We updated our manuscript on page 3 in Methods section to better reflect this.

Point 5: No minimum duration of adherence

Response 5: Due to the observational nature of these prospective studies, diet was assessed mostly by semi-quantitative food frequency questionnaires, which reflects habitual diet but did not specifically ask about the duration of the study participants’ diet habits. This being a limitation of our study, we revised our manuscript on page 15 to reflect this. “Most studies assessed diet during enrolment and it may not reflect the association of subsequent dietary changes and CVD risks.”

Point 6: Only one author extracted data.

Response 6: Both ZHG and HCC were involved in the screening and extraction of the data. We updated the manuscript to clarify this.

Point 7: Highlighted details not provided in PROSPERO trial registration (i.e., minor discrepancies from trial registration).

Response 7: We have reviewed and updated our PROSPERO trial contents to better clarify the methods that we have used in the current study.

Point 8: No rationale provided for "higher consumption".

Response 8: We have included references and updated our text to clarify the meaning of higher consumption of plant foods on page 3.

Point 9: Subgroup analysis was not provided apriori. Meta-regression was not defined apriori.

Response 9: We have reviewed and updated our PROSPERO trial contents to better clarify the methods that we have used in the current study. On page 3, we stated “Subgroup analyses were performed to investigate potential sources of heterogeneity, stratified by sex, country, follow up duration, mean age at baseline, methods of dietary classification, exposure ascertainment and study quality. Meta-regression was also undertaken for continuous variables of study characteristics, including the mean age and BMI at baseline, the proportion of smokers and the length of follow up.”

Point 10: R version not provided.

Response 10: We have updated the manuscript with the R version used.

Point 11: Incorrect phrasing. “Although mean age at baseline, outcome definition, study region and study quality are significant sources of heterogeneity for the association between plant-based dietary pattern and CVD, CHD and stroke, the direction of association and significance of plant-based dietary pattern adherence on CVD and CHD risks were generally consistent across strata.”

Response 11: We have rephrased the paragraph as followed, “We found that the mean age at baseline, outcome definition, study region and study quality are significant sources of heterogeneity for the association between plant-based dietary pattern and the risk of CVD, CHD and stroke. Despite of the existence of heterogeneity, the direction of association and significant findings of adherence to plant-based dietary pattern on CVD and CHD risks were generally consistent across different strata.”

Point 12: Link does not work.

Response 12: We believe that the link will be updated by the editorial office once it is published.

Thank you again for your time, we have revised the manuscript and we look forward to hearing back from you.

Round 2

Reviewer 2 Report

Dear authors,

congrats to this much improved paper; however, what I miss is your AW to my general comments previously mentioned in round 1 and how you adressed these; and what is uncomfortable is what you changed and updated in PROSPERO but do not provide here in the AW to reviewers - pls. add (sometimes pages are not correctly mentioned so hard to find your corrections/adaptations, as you also did not colour it for easier detection).

I guess after I concrntrated final round with adequate adatations and AWs, it might be a review to be considered for publication.

All good success

Author Response

Response to Reviewer 2 Comments

Thank you again, we appreciate your comments and we believe that it will help us further improve our manuscript. We have updated our response and manuscript with additional clarification on the location of the changes we made. We have also attached the PROSPERO registration document for your reference.

Point 1: This work is not only a literature search but a systematic review and should be identified as such in the abstract. No details on inclusion/exclusion criteria provided. No details provided on risk of bias assessment. No brief summary on limitations provided. Source of funding not indicated. Register and register number not indicated.

Response 1: Being mindful of the limited words allowed (200 words in the abstract), we updated the abstract the best we can, to both highlight the main methods and results of this study, and to give relevant information regarding inclusion/exclusion criteria and registration number. Risk of bias and limitations of the study were included in the main text due to word limitations.

Point 2: What the background is lacking is an overview of randomized controlled trial (RCT) evidence in the area of the research investigation, as such evidence has the lowest risk of bias, followed by quasi-experimental evidence, and then observational evidence. The rationale is missing for why this review does not include RCT data or even experimental data but focuses entirely on observational data.

Response 2: We updated the manuscript to clarify our rationale of including observational studies. On page 1, line 42-46, we stated “Although randomized controlled trials involving plant-based diets have been shown to improve cardiometabolic risk factors, atherosclerosis is a slowly progressing disease. Prospective observational studies are preferable to reflect the associations of long-term dietary patterns on CVD risk.[3].”

Point 3: So, what is the goal after identifying the gap in the literature? Clearly outline what the aim of this specific review is. is NOT the same as describing in 1 sentence what you did

Response 3: We revised our introduction to better reflect our study aim. On page 2, line 66-77, we stated “Given the possible benefits and low risk for the adoption of plant-based diets on lowering CVD risk, a quantitative assessment of current research is warranted to provide conclusive evidence in informing clinical and public health recommendations. In addition, it is important to examine the association of vegetarian or plant-based diets on CHD and stroke separately since diet may affect them differently. In light of recent findings from several large prospective cohort studies, we conducted a systematic review and meta-analysis to address the knowledge gap regarding the associations between plant-based dietary patterns and total CVD, CHD and stroke incidence. In addition, we investigated the dose-response associations of adherence to PDI patterns and the risk of total CVD, CHD and stroke.”

Point 4: Population unclear: no age provided and no clear indication of baseline health status.

Response 4: In our literature search, we included studies with a population of healthy adults, excluding children and adults with previous cardiovascular disease diagnoses (Supplemental Table 2). We updated our manuscript on page 3, line 95-96 in Methods section to better reflect this.

Point 5: No minimum duration of adherence

Response 5: Due to the observational nature of these prospective studies, diet was assessed mostly by semi-quantitative food frequency questionnaires, which reflects habitual diet but did not specifically ask about the duration of the study participants’ diet habits. This being a limitation of our study, we revised our manuscript on page 18 to reflect this. “Most studies assessed diet during enrolment and it may not reflect the association of subsequent dietary changes and CVD risks.”

Point 6: Only one author extracted data.

Response 6: Both ZHG and HCC were involved in the screening and extraction of the data. We updated the manuscript in page 3, line 106, to clarify this.

Point 7: Highlighted details not provided in PROSPERO trial registration (i.e., minor discrepancies from trial registration).

Response 7: We have reviewed and updated our PROSPERO trial contents to better clarify the methods that we have used in the current study. We have attached the PROSPERO trial document as a reference.

Point 8: No rationale provided for "higher consumption".

Response 8: We have included references and updated our text to clarify the meaning of higher consumption of plant foods on page 3, line 118.

Point 9: Subgroup analysis was not provided apriori. Meta-regression was not defined apriori.

Response 9: We have reviewed and updated our PROSPERO trial contents to better clarify the methods that we have used in the current study. On page 4, line 154-159, we stated “Subgroup analyses were performed to investigate potential sources of heterogeneity, stratified by sex, country, follow up duration, mean age at baseline, methods of dietary classification, exposure ascertainment and study quality. Meta-regression was also undertaken for continuous variables of study characteristics, including the mean age and BMI at baseline, the proportion of smokers and the length of follow up.”

Point 10: R version not provided.

Response 10: We have updated the manuscript with the R version used in page, line 180.

Point 11: Incorrect phrasing. “Although mean age at baseline, outcome definition, study region and study quality are significant sources of heterogeneity for the association between plant-based dietary pattern and CVD, CHD and stroke, the direction of association and significance of plant-based dietary pattern adherence on CVD and CHD risks were generally consistent across strata.”

Response 11: We have rephrased the paragraph in page 16 as followed, “We found that the mean age at baseline, outcome definition, study region and study quality are significant sources of heterogeneity for the association between plant-based dietary pattern and the risk of CVD, CHD and stroke. Despite of the existence of heterogeneity, the direction of association and significant findings of adherence to plant-based dietary pattern on CVD and CHD risks were generally consistent across different strata.”

Point 12: Link does not work.

Response 12: We believe that the link will be updated by the editorial office once it is published.

Response to general comments: We thank you for giving us generous feedback. We updated our introduction to better introduce the research gap (lack of quantitative assessment of the association of plant-based diets and CVD, CHD, stroke incidence), the research question (are plant-based diets associated with lower CVD/CHD/stroke incidence? Is there a dose-response relationship?), and the studies that should be included to answer these questions (prospective observational studies that investigate the association of plant-based diets and CVD/CHD/stroke incidence).

In the first paragraph of our discussion, we summarized the main findings of our results. 1) plant-based diets are associated with a lower risk of incident CVD and CHD, 2) significant dose-response association between adherence to the overall PDI and healthful PDI with lower CVD risk, 3) adherence to the unhealthful PDI was associated with higher CVD risk. Since the association of plant-based diets and stroke incidence was not significant, we chose to present and discuss it at the last paragraph of our discussion.

Thank you again for your time, we have revised the manuscript and we look forward to hearing back from you.

Thank you again for your time, we have revised the manuscript and we look forward to hearing back from you.
